# TRAIL-Based Therapies Efficacy in Pediatric Bone Tumors Models Is Modulated by TRAIL Non-Apoptotic Pathway Activation via RIPK1 Recruitment

**DOI:** 10.3390/cancers14225627

**Published:** 2022-11-16

**Authors:** Régis Brion, Malika Gantier, Kevin Biteau, Julien Taurelle, Bénédicte Brounais-Le Royer, Franck Verrecchia, Françoise Rédini, Romain Guiho

**Affiliations:** 1Nantes Université, Univ Angers, CHU Nantes, INSERM, CNRS, CRCI2NA, Team 9 CHILD, F-44000 Nantes, France; 2CHU Nantes, F-44000 Nantes, France; 3Nantes Université, CHU Nantes, INSERM, Bone Sarcomas and Remodeling of Calcified Tissues, UMR 1238, F-44000 Nantes, France; 4Nantes Université, CHU Nantes, INSERM, Center for Research in Transplantation and Translational Immunology, UMR 1064, F-44000 Nantes, France; 5Nantes Université, Oniris, CHU Nantes, INSERM, Regenerative Medicine and Skeleton, RMeS, UMR 1229, F-44000 Nantes, France

**Keywords:** TNF-related apoptosis inducing ligand (TRAIL), TRAIL-based therapies, osteosarcoma, Ewing sarcoma, pediatric bone tumor, resistance, non-apoptotic pathways

## Abstract

**Simple Summary:**

Osteosarcoma and Ewing sarcoma are the two most common malignant primary bone tumors occurring during childhood. These highly aggressive cancers are still clinically challenging, especially for recurrent/metastatic forms. Finding new therapeutic possibilities is critical to prolong survival of high-risk patients. One of the explored avenues is the exploitation of a tumor-specific vulnerability: the sensitivity to the pro-apoptotic cytokine TRAIL. However, we have previously depicted resistance mechanisms mitigating the effects of TRAIL, and TRAIL-mimicking strategies, in pediatric bone tumor models. Here, we describe one of these mechanisms, the activation of an alternative signaling pathway, alongside the apoptotic pathway, inducing pro-survival and even pro-proliferative processes. Then, we propose two different approaches to overcome this resistance mechanism: a proof-of-concept by inhibition of a key player of this non-apoptotic pathway; and a new compound able to increase the pro-apoptotic signal.

**Abstract:**

Despite advances in clinical management, osteosarcoma and Ewing sarcoma, the two most frequent malignant primary bone tumors at pediatric age, still have a poor prognosis for high-risk patients (i.e., relapsed or metastatic disease). Triggering a TRAIL pro-apoptotic pathway represents a promising therapeutic approach, but previous studies have described resistance mechanisms that could explain the declining interest of such an approach in clinical trials. In this study, eight relevant human cell lines were used to represent the heterogeneity of the response to the TRAIL pro-apoptotic effect in pediatric bone tumors and two cell-derived xenograft models were developed, originating from a sensitive and a resistant cell line. The DR5 agonist antibody AMG655 (Conatumumab) was selected as an example of TRAIL-based therapy. In both TRAIL-sensitive and TRAIL-resistant cell lines, two signaling pathways were activated following AMG655 treatment, the canonical extrinsic apoptotic pathway and a non-apoptotic pathway, involving the recruitment of RIPK1 on the DR5 protein complex, activating both pro-survival and pro-proliferative effectors. However, the resulting balance of these two pathways was different, leading to apoptosis only in sensitive cells. In vivo, AMG655 treatment reduced tumor development of the sensitive model but accelerated tumor growth of the resistant one. We proposed two independent strategies to overcome this issue: (1) a proof-of-concept targeting of RIPK1 by shRNA approach and (2) the use of a novel highly-potent TRAIL-receptor agonist; both shifting the balance in favor of apoptosis. These observations are paving the way to resurrect TRAIL-based therapies in pediatric bone tumors to help predict the response to treatment, and propose a relevant adjuvant strategy for future therapeutic development.

## 1. Introduction

The two most frequent pediatric malignant bone tumors, namely osteosarcoma and Ewing sarcoma, are rare forms of cancers, highly metastatic and cause serious disruption of bone remodeling. Osteosarcoma incidence peaks during the second decade of life coinciding with adolescent growth but its pathogenesis is not clearly determined [1]. Ewing sarcoma is identified by a chromosomal translocation leading to fusion of the EWS gene and a member of the ETS transcription factors family, in most cases FLI1, leading to the production of the fusion protein EWS-FLI1. This fusion protein acts as an aberrant transcription factor, directly or indirectly modulating the expression of numerous genes altering several cell functions such as cell cycle or apoptosis [2]. Although both tumors have distinct origins and pathogenesis, the clinical description and therapeutic approaches are similar: surgery, neoadjuvant and adjuvant chemotherapy, and radiotherapy for Ewing sarcoma only. The 5-year survival rate has barely changed over the past 30 years, remaining at around 70–75% for localized forms; however, this falls to 20% in metastatic or relapsed disease. Therefore, there is an urgent need for alternative therapeutic options, especially for these high-risk patients [3,4].

The role of TNF-related apoptosis inducing ligand (TRAIL), a pro-apoptotic cytokine from the TNF superfamily, has been widely studied in several cancer types for its ability to specifically target tumor cells. TRAIL is physiologically expressed by the innate immune system for anti-tumor surveillance [5], and can bind to five receptors: two death receptors TRAIL-R1 (Death Receptor 4: DR4) and TRAIL-R2 (DR5), and three decoy receptors that do not transmit the death signal but could confer resistance toward TRAIL-induced apoptosis: TRAIL-R3 (Decoy Receptor 1: DcR1), TRAIL-R4 (DcR2) and the soluble osteoprotegerin (OPG). TRAIL, by binding to DR4 and DR5, is able to induce apoptosis of tumor cells by recruiting the intracellular death domains of Fas-associated with death domain (FADD) protein, therefore activating the recruitment and cleavage of the pro-caspase-8 in a complex called a death-inducing signaling complex (DISC). Then, a recruitment cascade leads to tumor cell death through the activation of the extrinsic apoptosis pathway, which can be amplified in some cases by the activation of the intrinsic apoptosis pathway [6], both leading to the final cleavage of pro-caspase-3 into active caspase-3 [7]. However, the binding of TRAIL or its agonists on death receptors has also been linked to non-apoptotic signaling pathways induction, such as NF-κB [8], JNK [9], p38 [10], MAPK superfamily [11,12,13], Src, Rac1 [6], and Akt [14]; therefore, stimulating the tumor cell survival, proliferation and/or migration.

The commitment between these apoptotic or non-apoptotic pathways involves early complex molecular events that are still only partially understood. Concerning the activation of the non-apoptotic pathway, co-immunoprecipitation experiments revealed the formation of a secondary signaling complex after the DISC complex assembly, composed of RIPK1 (receptor-interacting protein kinase 1), TRAF2, NEMO (NF-κB essential modulator), FADD and active caspase-8. Silencing strategies of these different actors have demonstrated the requirement of RIPK1 and TRAF2 as an adaptor protein for the assembly of this secondary signaling complex [15,16]. A modified DISC complex has been involved in TRAIL resistance in non-small cell lung carcinoma, hepatic cancer, prostate cancer, leukemia, or glioblastoma models [6]. Interestingly, RIPK1 mediates the assembly of the DISC complex in the non-lipid raft fraction of the plasma membrane, thus leading to the inhibition of caspase-8 cleavage and NF-κB activation, subsequently inducing TRAIL resistance.

In the early 2000s, it was reported that several cell lines derived from Ewing sarcoma are sensitive to TRAIL-induced apoptosis in vitro [17,18,19], whereas osteosarcoma cell lines are mostly resistant to TRAIL [20]. In vivo, TRAIL and TRAIL-based strategies significantly reduce tumor volume of xenograft mouse models of Ewing sarcoma, prevent osteolytic lesions, and increase animal survival [21,22,23]. In addition, two publications reported a significant inhibitory effect of TRAIL overexpression in an allograft [22] and in a xenograft [24] mouse models of osteosarcoma. However, the pediatric bone tumor microenvironment also seems to promote TRAIL resistance [25].

To better understand the inconsistency of TRAIL-based therapies in osteosarcoma and Ewing sarcoma models, the objective of the present work was to explore the sensitivity of pediatric bone tumor cell lines and cell-derived xenograft models to TRAIL-based therapy, with a special emphasis on the induction of non-apoptotic pathways. AMG655/Conatumumab, a fully humanized monoclonal DR5 agonist antibody, was chosen as an example of TRAIL-based therapy [26]. Then, to define new options to overcome TRAIL resistance in pediatric bone sarcomas, we developed two independent strategies:
A proof-of-concept approach using gene silencing in order to decrease the formation of the secondary signaling complex.An innovative method to enhance pro-apoptotic signaling by increasing death receptors clustering. This strategy was based on previous promising experiments using mesenchymal stromal cells (MSCs) modified to express membraneous TRAIL [27,28,29]. In those studies, a large clustering of death receptors allowed a very efficient apoptosis of adjacent tumor cells. However, the use of these cells in clinical settings is highly limited because MSCs appear to exert a pro-tumor potential by themselves [23,25]. In this context, advanced tools able to induce the clustering of TRAIL death receptors while being neutral in the tumor microenvironment were developed, such as new TRAIL death receptor agonists able to bind six receptors [30,31,32].Here, we propose to test both strategies in the context of TRAIL-resistant osteosarcoma in vitro and in vivo.

## 2. Materials and Methods

### 2.1. Tumor Cell Lines

Four human Ewing sarcoma cell lines: SK-ES-1 (kindly provided by Dr. S. Burchill, Children’s Hospital, Leeds, UK), EW-24, A-673 and TC-71 (kindly provided by Dr. O. Delattre, INSERM U830, Paris, France), and four human osteosarcoma cell lines: K-HOS, SaOS-2, U-2 OS, and MG-63 (purchased from the ATCC) were used. All cell lines except TC-71 were cultured in Dulbecco’s Modified Eagle’s Medium (DMEM, Lonza, Basel, Switzerland) with 10% fetal bovine serum (FBS; Eurobio Sci., Les Ulis, France) and 2 mmol/L L-glutamine. TC-71 cells were cultured in Rosewell Park Millenium Institute (RPMI, Lonza) medium with 5% FBS and 2 mmol/L L-glutamine. 

### 2.2. Cell Viability

Two thousand cells per well were seeded into 96-well plates and cultured for 72 h in the presence of increasing concentrations of recombinant human TRAIL (R&D systems, Minneapolis, USA), AMG655 (Amgen, Thousand Oaks, CA, USA) or APG880 (Apogenix, Heidelberg, Germany) as indicated. Cell viability was determined by Cristal Violet staining (Sigma-Aldrich, St. Louis, MO, USA) as previously described [33]. IC50 were calculated using GraphPad Prism v9.4.0 software modelization. Molecular weights are reported in Appendix A.

### 2.3. Caspase-3/7 Enzymatic Activity

Caspase-3/7 activity was measured using an Apo-ONE^®^ Homogeneous assay kit (Promega, Madison, WI, USA). Briefly, cells were lysed in RIPA buffer (10 mM Tris pH8, 1 mM EDTA, 150 mM NaCl, 1% NP40, 0.1% SDS) containing a cocktail of protease and phosphatase inhibitors (1 mM Na_2_VO_4_, 1 mM phenylmethylsulforyl fluoride, Protease Cocktail Inhibitor, Sigma) at 4 °C. An equal volume of reagents was added to a black-walled, black bottom 96-well plate containing protein extracts and incubated at room temperature for 16 h. The fluorescence of each sample was measured in a plate-reader (Tristar LB941, Berthold technologies, Bad Wildbad, Germany). Caspase-3/7 activity was normalized to total protein concentration, determined by BCA kit (Sigma) minus basal activity of untreated cells.

### 2.4. Western Blot Analysis

Cells were lysed in RIPA buffer and the protein concentration was determined as described above. Samples containing 60 µg of total protein extracts in Laemmli buffer were separated by SDS-polyacrylamide gel electrophoresis and transferred to polyvinylidene fluoride Transfer membrane (PVDF, Immobilon FL, Millipore, Burlington, VT, USA). The membranes were blocked in Odyssey Blocking Buffer (LI-COR Bioscience, Lincoln, NE, USA) at room temperature for 1 h and blots were probed overnight at 4 °C with primary antibodies (list in Appendix A). Membranes were then incubated for 1 h with 1:10,000 diluted secondary fluorescent antibodies (LI-COR Bioscience) at room temperature. Specific proteins were detected using Odyssey^®^ Fc (LI-COR Bioscience) after washing. Western blot were analyzed with Image Studio Lite software (LI-COR Bioscience). 

### 2.5. Immunoprecipitation

For immunoprecipitation, K-HOS or TC-71 cells were treated with 5 μg⋅mL^−1^ AMG655 for 30 min before cell lysis in Pierce IP lysis buffer (Thermo Fisher Scientific, Waltham, MA, USA) supplemented with protease cocktail inhibitor. One mg of total protein was incubated with AMG655 antibody, protein A and protein G microbeads (Miltenyi Biotec, Bergisch Gladbach, Germany) for 2 h on solar agitation at 4 °C. The mix was added to μMACS column, rinsed 4 times with lysis buffer then 1 time with low salt wash buffer and finally eluted accordingly to the manufacturer’s instructions. The immune complexes were analyzed by Western blot using rabbit anti-RIP and anti-DR5 antibodies (Cell Signaling Technology, Danvers, MA, USA).

### 2.6. RNA Interference

The mRNA of RIPK1 gene was targeted using an RNA interference strategy. Three sequences were tested to select the one inducing the strongest inhibition of the target gene expression. shRNA products were named shRIPK1-1 (ATACCACTAGTCTGACGGA), shRIPK1-2 (CGGAACAGATTCTGGTGTCTT-Sigma SHCLND-NM_003804), and shRIPK1-3 (GCAGTCTTCAGCCCATTAAAT-Sigma SHCLND-NM_003804). Vectors encoding shRNAs were incorporated into the tumor cell genome by lentiviral strategy using a protocol previously described [34].

### 2.7. In Vivo Experiments of Ewing Sarcoma and Osteosarcoma Preclinical Models

Four-week-old Rj:NMRI-nude mice (Janvier labs—Le Genest-Saint-Isle, France) were housed under pathogen-free conditions at the Experimental Therapeutics Unit (Medical school, Nantes, France). The mice were anesthetized by inhalation of an isoflurane/air mixture before receiving an intramuscular injection of 1 × 10^6^ TC-71 Ewing sarcoma cells or K-HOS osteosarcoma cells in close proximity to the tibia. The tumor volume (V) was calculated from the measurement of three perpendicular diameters using a caliper, according to the following formula: V = 0.5 × length × width × height. Mice were sacrificed when tumor volume reached 2000 mm^3^ for ethical reason. For consistency across the experiments shown here, the arbitrary limit of 1200 mm^3^ was assigned for Kaplan–Meier estimator plots.

### 2.8. Statistical Analyses

Each experiment was repeated independently 3 times. Results are given as a mean ± SD for in vitro experiments and mean ± SEM for in vivo experiments. They were compared using unpaired t test or two-way ANOVA followed, respectively, by Bonferroni’s post-test or Tukey’s multiple comparisons test, calculated with GraphPad Prism v9.4.0 software. Results with *p* < 0.05 were considered significant and represented by an asterisk (*), *p* < 0.01 represented by two asterisks (**), and *p* < 0.001 represented by three asterisks (***).

## 3. Results

### 3.1. Osteosarcoma and Ewing Sarcoma Cell Lines Display Discrepancy in Sensitivity to the Pro-Apoptotic Effect of rhTRAIL and of the DR5-Agonist AMG655

The TRAIL-sensitivity of Ewing sarcoma (EWS) and osteosarcoma (OS) cell lines have been thoroughly documented [20]; we completed this study with the simultaneous assessment of TRAIL-sensitivity and the sensitivity to DR5-agonist AMG655. Crystal violet proliferation assays in the presence of increasing concentrations of rhTRAIL or AMG655 allowed us to determine IC50 for each cell line at 72 h post-treatment when the resulting differences between conditions are magnified (Figure 1A and Appendix A). Both EWS and OS cell lines presented different but correlated sensitivities to rhTRAIL and AMG655, as illustrated by the examples TC-71 (EWS) and K-HOS (OS) cell lines in Figure 1B,C. TC-71 is an example of a highly sensitive cell line to the pro-apoptotic effect of both rhTRAIL and AMG655, with an IC50 < 150 ng⋅mL^−1^ (IC50 < 1 nM) associated with a significant and persistent caspase-3 activity at 8 h, 16 h, and 24 h of treatment (Figure 1D,E). K-HOS cell line is an example of a TRAIL-resistant and AMG655-resistant cell line, with an IC50 > 1000 ng⋅mL^−1^ (IC50 > 10 nM), associated with a weak caspase-3 activity. In order to decipher the early apoptosis induction in these two cell lines, we performed a Western blot analysis of three markers: cleavages of caspase-8, caspase-3, and Poly [ADP-ribose] polymerase 1 (PARP-1) (Figure 1F). Both cell lines showed an early caspase-8 cleavage (18 kDa band), from 1 h to 6 h post AMG655-treatment, with a peak at 2 h, compatible with an extrinsic apoptosis pathway induction. This was associated with an important and sustained cleavage of caspase-3 (17 kDa band) and PARP-1 (89 kDa band) only in TC-71 cell line, which could indicate an amplification of the signal by the induction of the intrinsic mitochondria-dependent apoptosis pathway, whereas the cleavages were slight and only transient in K-HOS. Together, our data suggest that although both cell lines are able to engage the extrinsic apoptosis pathway following rhTRAIL or AMG655 treatments, only the K-HOS cell line could lessen this apoptosis induction.

### 3.2. In Vivo, AMG655 Slows down the Primary Tumor Growth in the Sensitive Model Induced by TC-71 Cells but Accelerates the Tumor Growth in the Model Induced by K-HOS Resistant Cells

Because TC-71 cells were shown to be sensitive to the TRAIL death receptor agonist AMG655, a para-tibial orthotopic tumor model was developed by injection of these cells to confirm, or not, the effects of AMG655 in vivo. Four groups of six mice were injected with TC-71 cells on day 0 (D0). Tumors became palpable at D7 and treatment with AMG655 started at D1. Each group was intravenously injected twice a week with NaCl 0.9% (vehicle) or 0.2; 2 or 4 mg⋅kg^−1^ of AMG655. All treated groups exhibited a significant reduction of the tumor growth starting from D15 (Figure 2A). At D22, the group injected with 0.2 mg⋅kg^−1^ AMG655 presented a 40% reduced mean tumor volume, whereas the mean volume of groups treated with 2 or 4 mg⋅kg^−1^ was decreased by 60% compared to the control group. However, all treated mice developed tumors, meaning that AMG655 injection was not sufficient to completely abolish tumor occurrence. Nevertheless, the survival rate determined when the tumor volume reached 1200 mm^3^ was significantly higher in the treated groups (Figure 2B) with the best prognosis for the group treated at the dose of 2 mg⋅kg^−1^ AMG655. This dose was selected for all subsequent experiments. 

Since the K-HOS cell line was found resistant to AMG655 in vitro, it was used for the induction of an orthotopic para-tibial tumor model resistant to TRAIL-based therapy. Two groups of seven mice were injected with one million K-HOS cells at day 0 (D0), the tumors became palpable at D8, and the treatment with AMG655 started at D1. Each group was injected twice a week with vehicle or with 2 mg⋅kg^−1^ of AMG655. Unexpectedly for a resistant model, instead of a slight decrease or an absence of response to DR5 agonist, the group injected with AMG655 presented an accelerated tumor growth with a tumor volume significantly higher at D30, and up to 30% higher compared to the control group at D37 (Figure 2C). As a consequence, the survival rate of the group injected with AMG655 was significantly decreased as compared to the control group (Figure 2D). 

### 3.3. TRAIL Non-Apoptotic Pathways Are Activated by AMG655 in K-HOS Cells

To understand the pro-tumoral effect of DR5 agonist observed in the cell-derived xenograft model induced by K-HOS, we hypothesized that an activation of TRAIL non-apoptotic pathways could occur following AMG655 binding on DR5. A Western blot analysis of phospho-IκBα and phospho-c-Jun was used to monitor the NEMO/NF-κB and MAPK activations through the JNK pathways (Figure 3A). K-HOS and TC-71 cells were starved for 16 h, then treated with 1 µg⋅mL^−1^ of AMG655 for up to 2 h, corresponding to the described caspase-8 cleavage timeframe. We revealed a strong and sustained increase of IκBα phosphorylation in K-HOS, whereas IκBα phosphorylation in TC-71 was delayed and weaker. Accordingly, the phosphorylated form of c-Jun was boosted in K-HOS cells after 2 h of treatment while it remained stable in TC-71 cells. These results suggest that the non-apoptotic pathway was activated in both TRAIL-sensitive and TRAIL-resistant cells following DR5 engagement, but in a much stronger and sustained manner in the latter case.

RIPK1 recruitment to the secondary signaling complex has been found to represent a critical component of TRAIL non-apoptotic signaling. This has even been shown to promote cell survival, proliferation, and migration [35]. Therefore, we performed an immunoprecipitation using the AMG655 DR5-agonist antibody to assess the recruitment of RIP1 K in the protein complex (Figure 3B). In TC-71 cells, RIPK1 was not detected in the complex before treatment with AMG655 but was slightly engaged after 30 min. In the K-HOS cell line, RIPK1 was present in a relatively small amount before and recruited abundantly following 30 min of treatment. The negative control fraction is shown in Appendix A.

### 3.4. The Inhibition of Non-Apoptotic Pathways by Secondary Complex Disruption through RIPK1 Knockdown Leads to an Increase Caspase-3 Cleavage and AMG655 Re-Sensitization in K-HOS Cells

To confirm the key role played by RIPK1, we employed a silencing strategy using shRNA integrated into the genome by lentiviral strategy. Among the three shRNA tested, shRIPK1-1 sequence was the most efficient to reduce RIPK1 transcript and protein expression (Appendix A). K-HOS cells expressing shLacZ were used as control. Kinetic analysis following treatment with 1 µg/mL of AMG655 revealed that although caspase-8 cleavage was not affected by RIPK1 knockdown, caspase-3 cleavage was increased as evidenced by the appearance of the 17 kDa band as early as 2 h, further magnified at 5 h following treatment (Figure 3C). This early caspase-3 cleavage attests an induction of the extrinsic apoptosis pathway. Simultaneously, RIPK1 knockdown reduced the phosphorylation of IκBα, particularly at 2 h post-treatment, and inhibited the phosphorylation of c-Jun in the K-HOS cell line (Figure 3D). Finally, proliferation assays in the presence of AMG655 confirmed that shRIPK1 expressing cells were sensitized to AMG655 compared to control K-HOS cells expressing shLacZ. A 50% inhibition of viability was indeed observed following the treatment with 500 ng⋅mL^−1^ of AMG655 (Figure 3E and Appendix A).

### 3.5. In Vivo, the Inhibition of RIPK1 Expression Is Able to Overcome K-HOS Cell Line AMG655-Resistance

The K-HOS shRIPK1-1 cell line, which showed the highest inhibition of RIPK1 expression and a significant blockade of the TRAIL non-apoptotic pathways activation following AMG655 treatment, was selected for the induction of para-tibial orthotopic osteosarcoma in nude mice. Six groups of seven mice were studied: two groups were injected intramuscularly either with K-HOS parental cells, K-HOS shLacZ cells, or K-HOS shRIPK1-1 cells. For each cell type, one group was injected twice a week with 0.9% NaCl (control group) and the other group with 2 mg⋅kg^−1^ of AMG655 (Figure 4 and Appendix A). Tumors induced by parental and shLacZ cells followed the same growth curve and were equally affected by the pro-tumoral impact of AMG655 (Appendix A). Vehicle-treated tumors induced by shRIPK1-1 and shLacZ cells showed similar growth curve, confirming that RIPK1-1 inhibition did not affect proliferation in vivo. However, only the mean volume of the shRIPK1-1 group was decreased by 50% when treated with 2 mg⋅kg^−1^ of AMG655 (Figure 4A). The survival rate was determined when the tumor volume reached 1200 mm^3^ and was significantly higher in the shRIPK1-1 AMG655-treated group compared to all the other groups (i.e., shRIPK1-1 vehicle-treated group and both treated and non-treated shLacZ groups) (Figure 4B).

Altogether, both in vitro and in vivo data indicate that an inhibition of TRAIL-non apoptotic pathways by RIPK1 knockdown is sufficient to re-sensitize K-HOS cells to AMG655.

### 3.6. APG880, a TRAIL-Death Receptors Multiple Agonist, Is Able to Induce Cell Death by Apoptosis in AMG655-Resistant K-HOS Cell Line 

We next investigated the potential of a novel TRAIL-death receptor agonist, APG880, composed of an assembly of two trimeric TRAIL chain molecules covalently fused with the Fc fragment of a human immunoglobulin. Thanks to this structure, each APG880 molecule is able to bind six TRAIL death receptors whereas AMG655, a complete immunoglobulin, can bind only two DR5 (Schematic representation in Figure 5A). Compared to AMG655, APG880 is, therefore, an ideal candidate to optimize death receptor clustering. We hypothesized that APG880 could bypass the induction of the TRAIL non-apoptotic signaling pathways and induce apoptosis of tumor cells. Of note, APG880 and AMG655 have a very similar molecular mass of about 150 kDa, and their effectiveness can be compared at the same concentration. Proliferation assay demonstrated that the K-HOS AMG655-resistant cell line (IC50 > 1000 ng⋅mL^−1^) was sensitive to APG880 with an IC50 < 50 ng⋅mL^−1^ (Figure 5B). This sensitization was highlighted by the activation of apoptosis response, attested by caspase-3/7 activity at 8 h, that persisted up to 24 h post APG880-treatment with concentrations higher than the IC50 (Figure 5C). We confirmed the cleavage of caspase-8, caspase-3, and PARP-1 by Western blot at 6 h (Figure 5D).

### 3.7. In Vivo, APG880 Inhibits Tumor Development in a K-HOS Xenograft Model

Based on the potent in vitro effect described previously, we developed a paratibial orthotopic K-HOS tumor model to investigate the impact of APG880 treatment in vivo. Four groups of 7 mice were injected with K-HOS cells at Day 0 (D0). Each group was injected twice a week with 0; 0.3; 1 or 3 mg⋅kg^−1^ of APG880. All treated groups presented a similar and significant inhibition of tumor progression, up to 80% observed at D39 (Figure 6A). This response was associated with a significant impact on survival rate (calculated when tumor volume reached 1200 mm^3^), which increased by 50% following treatment in the three groups (Figure 6B).

## 4. Discussion

Primary malignant bone tumors still represent a challenging clinical issue for pediatric oncologists, especially in the relapsed or metastatic setting. Finding new therapeutic strategies is, therefore, critical to offer more effective treatments for high-risk patients. Since the pro-apoptotic inducer TRAIL was discovered in the early 2000s, the research on TRAIL-based therapies has proven its relevance in the context of pediatric bone tumors, but has also highlighted several limitations such as treatment resistance [20]. First-generation clinical trials, including on AMG-655 (Conatumumab®), a fully humanized monoclonal antibody binding to and activating DR5, have been discouraged. Indeed, several phase 1/2 clinical trials have been completed, some involving patients with sarcomas (NCT00626704, NCT01327612), but never demonstrated remarkable improvement in patients’ survival [36]. In this study, to better understand the inconsistency of TRAIL-based therapies efficiency in osteosarcoma and Ewing sarcoma models, we investigated not only the induction of the classical apoptotic pathway but also alternative non-apoptotic pathways induced after the engagement of TRAIL death receptors.

Surprisingly, we highlighted that both pro- and non-apoptotic signaling pathways were induced in vitro after AMG655 treatment regardless of TRAIL sensitivity. Our experiments showing the activation of caspase-3 and PARP indeed suggest that even in resistant cell lines, the pro-apoptotic pathway is not impaired; however, further investigation could be implemented to finely detail the kinetic and to decipher cellular death mechanisms. Interestingly, although the activation of the pro-apoptotic pathway was sustained only in sensitive cells, the activation of the non-apoptotic pathway was stronger in resistant cells. Consequently, the resulting balance of these two pathways was different, leading to apoptosis only in sensitive cells. While the induction of the non-apoptotic pathway is already described in the literature, with regulatory steps extensively documented and reported in the comprehensive review by Azijli et al. [16], here, we demonstrated for the first time the resulting pro-tumoral effect of the AMG655 treatment in an osteosarcoma cell-derived xenograft model induced by resistant cells. Of note, this result also highlights the importance of taking into account the tumor microenvironment for TRAIL-based therapies studies. In that respect, in our in vitro settings, the activation of TRAIL death receptors by AMG655 was not able to increase the proliferation of the K-HOS cell line. The other way around, K-HOS cells migration was enhanced in vitro by AMG655, although the effect on lung metastasis in the xenograft model induced by K-HOS cells was not significant in our in vivo experimental conditions (data not shown). We hypothesize that osteosarcoma cells develop strategies to adapt to the immune system effector mechanisms of cancer surveillance, including TRAIL expression by NK and innate lymphoid cells. These cells could not only repress the signals that would lead to apoptosis in normal conditions, but also turn the signaling features to their advantage; therefore, promoting not only survival and proliferation, but also migration and invasion. The existence of these non-apoptotic signaling pathways, recently linked to nuclear translocation of TRAIL death receptors [37], and associated with pro-tumoral effects in several pre-clinical studies [38,39], including in the K-HOS osteosarcoma model herein described, calls for extra precaution in the clinical use of TRAIL death receptors activators.

In this study, we suggested two distinct types of sensitization strategies by: (1) acting on tumor cells themselves to modulate the intracellular signaling downstream the activation of death receptors using shRIPK1; and (2) using an innovative strategy (APG880) to increase the clustering of membrane death receptors in order to favor the apoptotic pathway instead of the proliferation and survival pathways.

First, targeting RIPK1, a key factor of the secondary signaling complex formation, using shRNA, allowed us to overcome resistance in the in vivo K-HOS model of osteosarcoma. We, here, provided important insights into: (1) the role played by the non-apoptotic pathways induced by TRAIL in the resistance of this model to the DR5-agonist AMG655; (2) the validation of TRAIL-based therapies in pediatric bone tumors; and (3) the proof of concept for new adjuvant targets that can synergize with TRAIL receptors agonists. Several synergies have already been described in the literature [40], here we can suggest, for example, Necrostatine-1 that inhibits RIPK1 kinase activity [41,42]. This compound has already demonstrated therapeutic interest thanks to its ability to inhibit TRAIL-induced necroptosis by acidic pH in liver and colon cancer cell lines [43]. Future studies should now determine whether Necrostatine-1 can inhibit the activity of the secondary TRAIL-signaling complex. However, it is important to note that the targeting of this pathway remains very hypothetical. Indeed, following the activation of the non-apoptotic TRAIL-induced pathway, RIPK1 seems involved as a scaffold protein and its kinase activity does not seem to be involved [44]. In addition, one should not overlook that RIPK1 is also involved in numerous signaling pathways and its systemic inactivation could, for example, inhibit TNF-α-induced apoptosis and lead to deleterious side-effects. As an alternative synergy opportunity, the modulation of specific miRNA could also be of interest. Indeed, miR-3132 has, for example, recently been shown to potentiate anti-proliferative and pro-apoptotic effects of TRAIL [45].

Our second strategy for successfully reversing the pro-tumoral effect of TRAIL-based therapy in a resistant osteosarcoma model took advantage of the second-generation TRAIL-receptor agonist APG-880 (ABBV-621-Eftozanermin). This innovative molecule has been developed to concomitantly bind up to six TRAIL receptors, consequently improving receptor clustering on tumor cells surface and boosting anti-tumor efficiency [30,31,32]. APG880 has already demonstrated anti-tumor activity in diffuse large B-cell lymphoma, acute myeloid leukemia, and multiple myeloma preclinical models [46], and a recently completed phase I clinical trial provided encouraging results with acceptable toxicity in patients with recurring solid and hematologic tumors (NCT03082209) [47]. Here, for the first time, we successfully demonstrated that APG880 could help control TRAIL-resistant osteosarcoma growth and, therefore, represents a relevant clinical therapeutic option. 

## 5. Conclusions

The present study revealed the key roles played by non-apoptotic pathways induced by the engagement of TRAIL death receptors in pediatric bone tumor models. For the first time, we described that the balance between pro- and non-apoptotic signals defines the outcome of the treatment. Further investigations will be necessary to better understand the apoptotic pathway kinetic and expend the list of actors involved in the non-apoptotic pathway. Moreover, the resistance to TRAIL-based therapies seems to be overcome using therapeutic strategies allowing a shift of this balance in favor of apoptosis. In order to test this hypothesis, we proposed two possible strategies consisting of the inhibition of the non-apoptotic pathway by targeting RIPK1 or the improved activation of the pro-apoptotic pathway using substantial death receptors clustering.

Further approaches integrating biomarkers to improve the prediction of the pro-/non-apoptotic signals balance will be needed to propose safer TRAIL-based therapies associated with a relevant adjuvant strategy to improve the management of pediatric bone tumors.

## Figures and Tables

**Figure 1 cancers-14-05627-f001:**
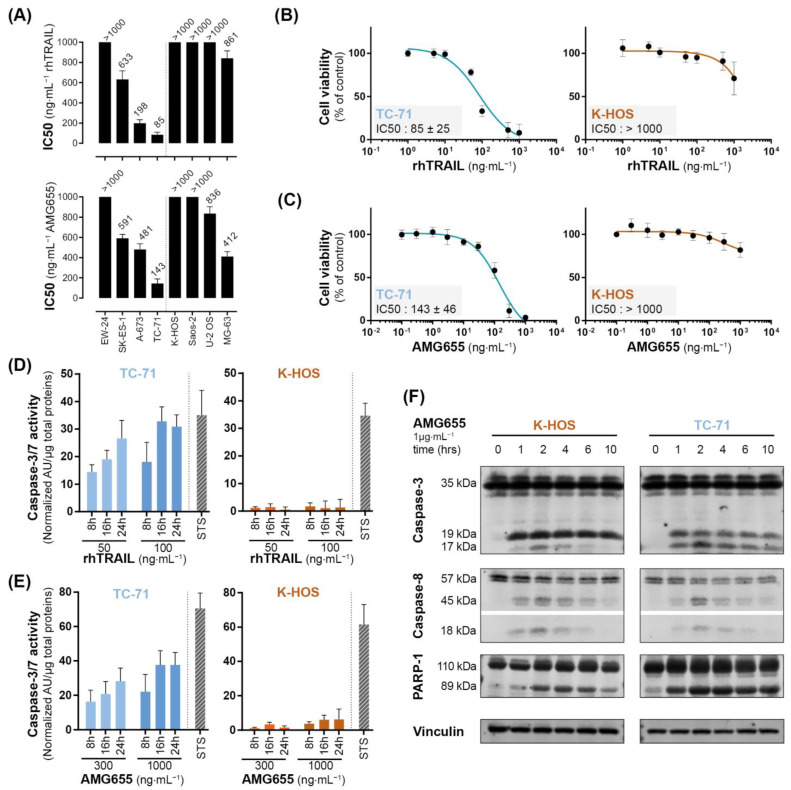
**Osteosarcoma and Ewing sarcoma cell lines show discrepancy in sensitivity to the pro-apoptotic effect of rhTRAIL and of the DR5-agonist AMG655.** (**A**) rhTRAIL and AMG655 IC50 of 4 osteosarcoma and 4 Ewing sarcoma cell lines. Cells were cultured for 72 h in the presence of increasing concentrations of rhTRAIL or AMG655; viability was determined by crystal violet staining. IC50 were calculated with GraphPad Prism software. As an illustration, viability curves of TC-71 (EWS) and K-HOS (OS) cell lines in the presence of (**B**) rhTRAIL or (**C**) AMG655 are shown. (**D**) Sensitivity to rhTRAIL- and (**E**) to AMG655-induced apoptosis was assessed by specific enzymatic assay of caspase-3/7 activity. Cell lines were treated with either 50 and 100 ng⋅mL^−1^ of rhTRAIL or 300 and 1000 ng⋅mL^−1^ of AMG655; total proteins were extracted 8 h, 16 h, or 24 h after treatment. The caspase-3/7 enzymatic activity was measured by Apo-ONE^®^ Homogeneous Caspase (Promega); the results were normalized to total protein. Staurosporine 1 µM (STS) treatment for 8 h was used as positive control of apoptosis induction. (**F**) Kinetics of caspase-8, caspase-3 and PARP-1 cleavages in K-HOS OS cell line and TC-71 EWS cell line following 1, 2, 4, 6, and 10 h of treatment with 1 µg⋅mL^−1^ AMG655 determined by Western blot. Vinculin was used as loading control.

**Figure 2 cancers-14-05627-f002:**
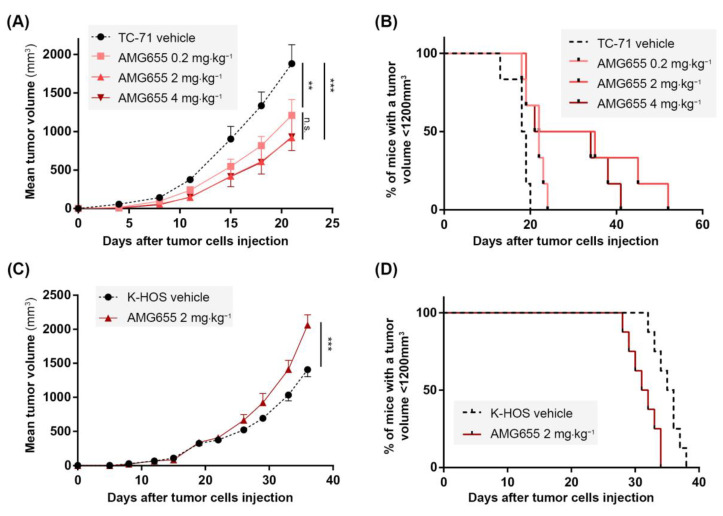
**In vivo, AMG655 slows down the primary tumor growth of the Ewing sarcoma model induced by TC-71 cells sensitive to rhTRAIL but accelerates the osteosarcoma model tumor progression induced by resistant K-HOS cells.** Four groups of six mice were injected with 1 × 10^6^ TC-71 cells as described in the “Materials and Methods” section. Mice of each group were injected twice a week with 0, 0.2, 2, or 4 mg⋅kg^−1^ of AMG655. (**A**) Average tumor volume per group. (**B**) Percentage of mice with a tumor volume below 1200 mm^3^ for the three groups. Two groups of seven mice were injected with 1 × 10^6^ K-HOS cells. Each group was injected twice a week with 0 or 2 mg⋅kg^−1^ of AMG655. (**C**) Average tumor volume per group. (**D**) Percentage of mice with a tumor volume below 1200 mm^3^ for the two groups. (** *p* < 0.01, *** *p* < 0.001).

**Figure 3 cancers-14-05627-f003:**
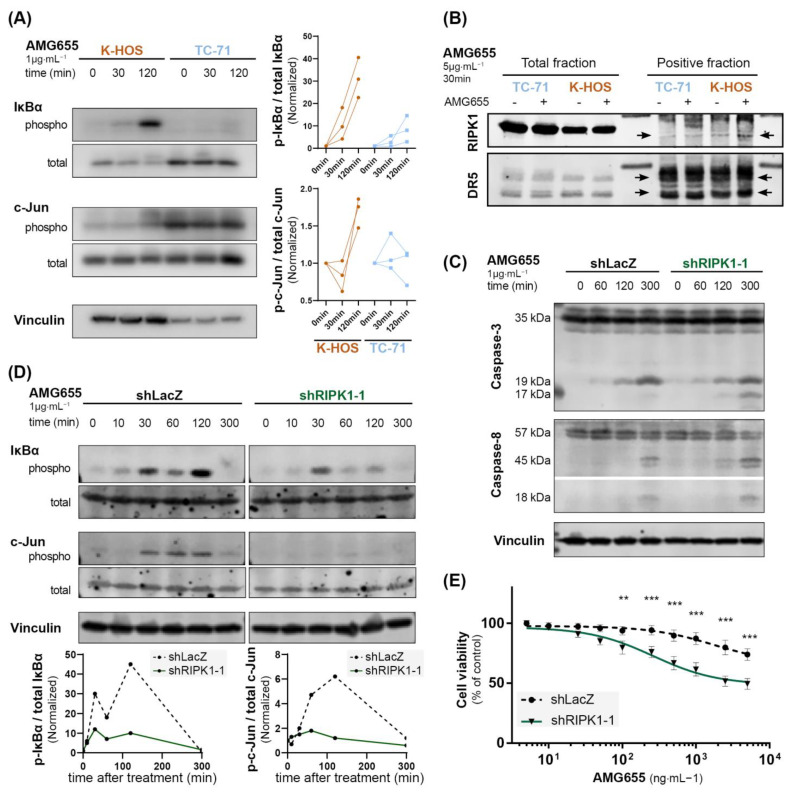
**The inhibition of non-apoptotic pathways in K-HOS cells by complex II disruption through RIPK1 knockdown leads to an increased caspase-3 cleavage and AMG655 re-sensitization.** (**A**) Kinetic of IκBα and c-Jun phosphorylation in K-HOS and TC-71 cell lines upon 1 µg⋅mL^−1^ of AMG655 treatment (left panels). Quantification of 3 independent experiments, normalized to total form and basal signal (right panels). (**B**) Immunoprecipitation of DR5 using AMG655 antibody to explore the recruitment of RIP1K in the protein complex. (**C**) Kinetics of caspase-8 and caspase-3 cleavages in K-HOS cell line transduced with shLacZ or shRIPK1-1 after 1, 2, and 5 h treatment with 1 µg⋅mL^−1^ of AMG655 analyzed by Western blot. Vinculin was used as a loading control. (**D**) Kinetic of IκBα and c-Jun phosphorylations in K-HOS shLacZ and shRIPK1-1 cell lines upon treatment with 1 µg⋅mL^−1^ of AMG655 (top panels) and quantification normalized to total form and basal signal (bottom panels). (**E**) shLacZ and shRIPK1-1 K-HOS cells were cultured for 72 h in the presence of increasing concentrations of AMG655 (** *p* < 0.01 *** *p* < 0.001).

**Figure 4 cancers-14-05627-f004:**
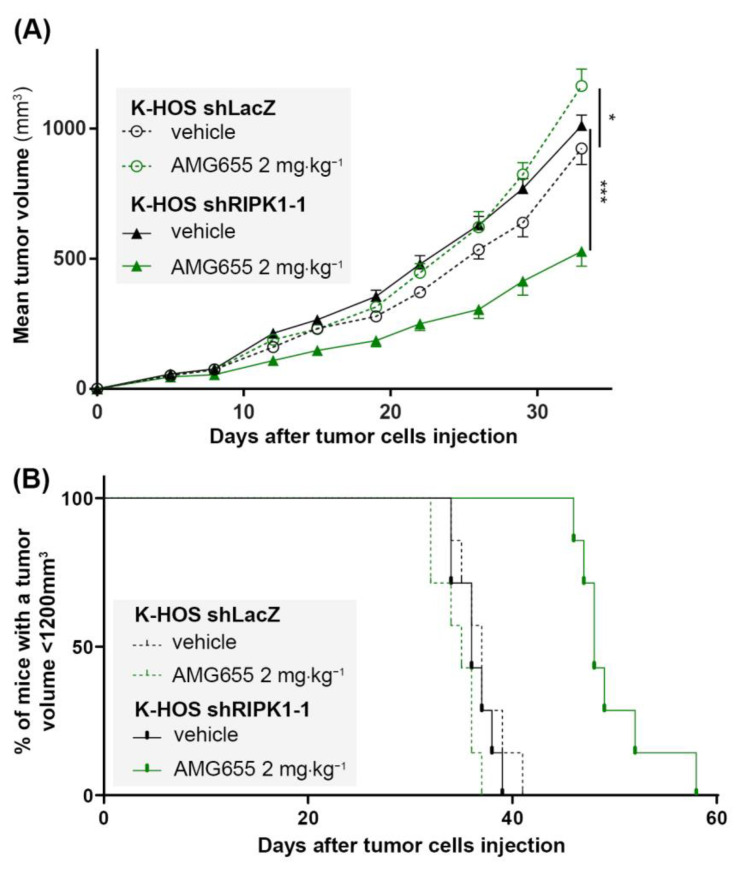
**The inhibition of RIPK1 expression by shRNA re-sensitizes K-HOS cell-induced tumors to the anti-tumor effect of AMG655.** Four groups of seven mice were studied: two groups received a paratibial injection of 1 × 10^6^ K-HOS shLacZ cells and two groups 1 × 10^6^ K-HOS shRIPK1-1 cells. For each type of cells, one group of mice was injected twice a week with 0.9% NaCl (control group) and the other group with 2 mg⋅kg^−1^ of AMG655. (**A**) Mean tumor volume per group calculated twice a week. (**B**) Survival evaluated by the percentage of mice with a tumor volume less than 1200 mm^3^. (* *p* < 0.01, *** *p* < 0.001).

**Figure 5 cancers-14-05627-f005:**
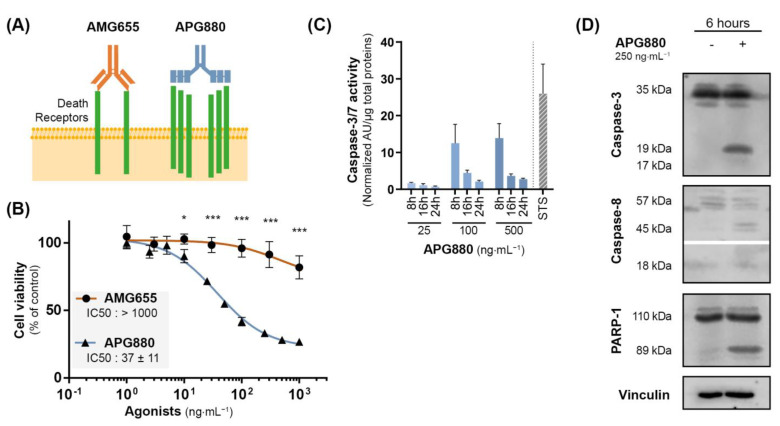
**APG880, a TRAIL-death receptors multiple agonist is able to induce apoptosis in K-HOS cells.** (**A**) Comparison between the immunoglobulin AMG655, and APG880 which consists of an assembly of two trimeric TRAIL chain molecules covalently fused with the Fc fragment of a human immunoglobulin. While AMG655 can only bind two DR5, APG880 has been designed to bind six TRAIL death receptors. (**B**) Cells were cultured for 72 h in the presence of increasing concentrations of AMG655 or APG880. IC50s were determined by crystal violet staining. (**C**) Sensitivity to APG880-induced apoptosis was assessed by specific enzymatic assay of caspase-3/7 activity. K-HOS cells were treated with either 25, 100, or 500 ng⋅mL ^−1^ of APG880, the cells were then lysed, and total proteins were extracted 8 h, 16 h, or 24 h following the treatment. The caspase-3/7 enzymatic activity was measured by Apo-ONE^®^ Homogeneous Caspase (Promega); the results were normalized to total protein. Staurosporine 1 µM (STS) for 8 h was used as positive control of apoptosis induction. (**D**) Kinetics of caspase-3, caspase-8, and PARP-1 cleavages in K-HOS cell line after 6 h of treatment with 250 ng⋅mL^−1^ APG880 was determined by Western blot. Vinculin was used as loading control. (* *p* < 0.01, *** *p* < 0.001).

**Figure 6 cancers-14-05627-f006:**
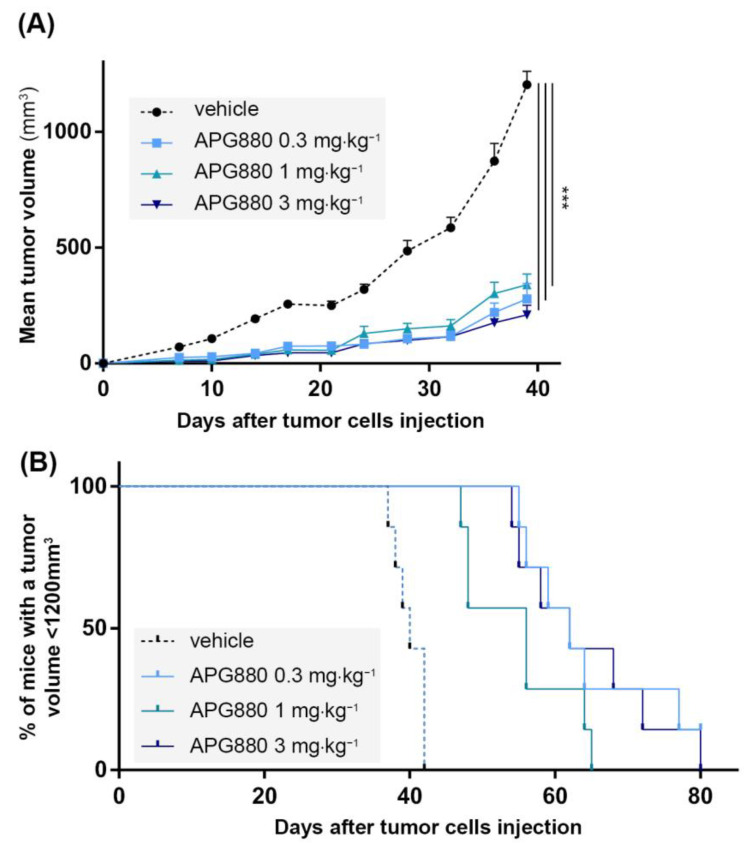
**In vivo, APG880 decreases K-HOS cell-induced tumor progression.** Four groups of seven mice received intramuscular paratibial injection of 1.10^6^ K-HOS cells. Each group was injected twice a week with 0; 0.3; 1 or 3 mg⋅kg^−1^ of APG880. (**A**) Mean tumor volume per group calculated twice a week. (**B**) Survival evaluated by the percentage of mice with a tumor volume less than 1200 mm^3^. (*** *p* < 0.001).

## Data Availability

Not applicable.

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
