# Peer review of "TRAIL-Based Therapies Efficacy in Pediatric Bone Tumors Models Is Modulated by TRAIL Non-Apoptotic Pathway Activation via RIPK1 Recruitment"

_cancers, 2022, doi:10.3390/cancers14225627_

Round 1
Reviewer 1 Report
The authors analyzed the TRAIL-based therapies efficacy in pediatric bone tumors osteosarcoma and Ewing sarcoma cell lines (four human OS cell lines, four human Ewing sarcoma cell lines) and two cell-derived xenograft models.
Tumour heterogeneity is taken into account by using four different cell lines in each case. The role of TRAIL, its binding possibilities to different receptors and the resulting apoptotic induction is well presented in the introduction.
A question about the study design: Why was RIPK1 selected as a target from the secondary signalling complexes?
Ad Fig1A-C: I would rather present the IC50 values of all cell lines in a table. The representation of IC50 values in ng/ml or µg/ml is unusual. A conversion (or the additional information) into nM and µM concentration ions would be better.
Ad Fig1D: Caspase 3/7 activity was measured after 8h, 16h and 24h. However, the IC50 values are from 72 h. When is the peak in caspase 3/7 activity reached? Were time points longer than 24 h also measured? While an increase in activity can be observed in the caspase 3/7 APO one assay, the western blots (Fig 1F) showed the strongest signals after 2-6 h already. Please explain these deviations. what do the results of the western blots look like after 24 or 72?
Ad Fig 2A/B and 2D/E: these results are reduntant
Ad Fig 3A: why were only IĸBα and c-Jun phoshorylation shown here? the IkB pathway consists of many more regulatory steps.
Ad Fig 3C: the caspase-3 and-8 kinetics were shown for 1-5 h. Is the peak already reached after 5 h? What do the results look like after e.g. 8 or 12 hours?
Ad Fig 4A/B: Here, too, the data are reduntant
Ad Fig 5B/C: If I understand correctly, APG880 has a highly significant lower IC50 concentration (37+11 ng/ml) than AMG655 (>1000 ng/ml). For the caspase 3/7 activity assay, similar high concentrations (100 and 300 ng/ml) were used as for AMG655 (300 and 1000 ng/ml). Due to the relatively much higher concentrations, the timing of the apoptosis maximum changes and is therefore difficult to compare.
Ad Fig 5D: Here, too, a time course would be interesting.
Ad Fig 6A/B: Here, too, the data are reduntant
The introduction is well developed and informative. The present study is methodologically sophisticated and quite well. However, the apoptotic induction - as the core point of the study - could still be supported with other methods.
Especially the in vivo data were repeatedly presented in reduntant figures. This is unnecessary as no additional knowledge gain is presented.
Seen in this light, I find the conclusion rather courageous and would formulate it somewhat more cautiously.
Reviewer 2 Report
Treatment of bone sarcoma is still a challenge and several approaches to improve treatment of relapsed and metastasized tumors have not been successful so far. The manuscript gives insight in the action of TRAIL signaling, highlighting that TRAIL inhibitors can induce both pro-apoptotic and anti-apoptotic pathways and the outcome depends on the balance of both in individual tumor cell lines. Apoptosis induction may be promoted by inhibitors with higher effectiveness and concomitant inhibition of anti-apoptotic pathways.
In future, a personalized approach evaluating specific targets in individual tumors and a multimodal therapy may improve outcome and increase survival.
The manuscript is comprehensible, well written and merits publication. There is only one minor point, which should be clarified prior to publication:
Minor
In chapter 2.7 you state that mice were sacrificed when tumor volume reached 2000 mm3 for ethical reason. In the results section survival is evaluated by the percentage of mice with a tumor volume less than 1200 mm3. Please explain the discrepancy.
Author Response
The authors appreciate the referee’s valuable comments, we are responding to the minor point raised bellow.
Point 1: The manuscript is comprehensible, well written and merits publication. There is only one minor point, which should be clarified prior to publication: In chapter 2.7 you state that mice were sacrificed when tumor volume reached 2000 mm3 for ethical reason. In the results section survival is evaluated by the percentage of mice with a tumor volume less than 1200 mm3. Please explain the discrepancy.
Response 1: The authors would like to thank the Reviewer 2 for noticing an apparent discrepancy in the material and method section. We confirmed that we have used the endpoint of 2000mm3 for the in vivo work presented in this study (Fig2;Fig4;Fig6 and FigS2), however, for clarity and consistency among the experiments, the arbitrary limit of 1 200 mm3 was assigned for Kaplan-Meier estimator plots. The method have been amended in the 2.7 section to specify this point (line 189).
Round 2
Reviewer 1 Report
My suggestions for revision from the review1 were largely accepted and adequately edited. Thus, I have no objections to publication of this manuscript.